# THE TWO-HOP CURSE:
# LLMS TRAINED ON $A{\to}B$, $B{\to}C$ FAIL TO LEARN $A{\to}C$

## ABSTRACT

While LLMs excel at answering multi-hop questions like "Who is the spouse of the performer of Imagine?" by thinking out loud (chain-of-thought), they perform surprisingly poorly when required to reason in their latent space and answer without chain-of-thought. This observation was previously referred to as the *compositionality gap*, implying that although language models are less reliable at two-hop latent reasoning, they still perform it sometimes. In this paper, we introduce a controlled setting for investigating the compositionality gap. We run a series of experiments finetuning a large language model (Llama-3-8B-Instruct) on synthetic facts expressed in English. We attempt to elicit two-hop reasoning in three ways: (i) fine-tune on a data mixture designed to incentivize two-hop reasoning, (ii) force facts to be stored in layers in the correct order, and (iii) use an auxiliary loss to provide activation-level supervision for two-hop reasoning. We show that LLaMA-3-8B successfully learns to answer two-hop questions about synthetic facts *using CoT*, but completely fails *without CoT*, achieving chance-level accuracy and chance-level test loss. Failures of LLMs in our controlled setting cast doubt on the purported ability of present LLMs to perform multihop latent reasoning and lead us to conjecture that, rather than a reasoning *gap*, current language models might exhibit a two-hop reasoning *curse* — a complete lack of ability rather than a relative weakness. This is the *Two-Hop Curse*.[1]

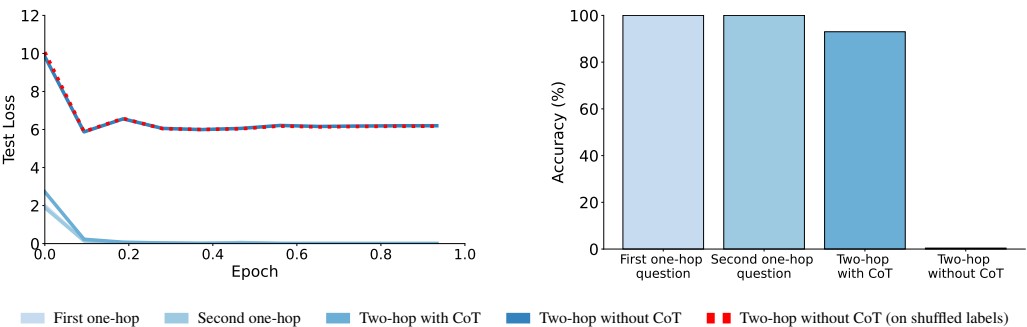

Figure 1: Performance of a baseline setup (described in detail in Section 4 on different question types (see Figure 2 for explanation). This model reaches perfect accuracy on one-hop questions and very high accuracy when giving CoT answers to two-hop questions but gets 0 accuracy without CoT — an example of the two-hop curse. None of our methods significantly improve upon this baseline.

## 1 INTRODUCTION

Large Language Models (LLMs) have shown remarkable reasoning abilities across a wide range of domains, particularly when prompted to think out loud (with chain-of-thought or CoT; Reynolds & McDonell, 2021; Wei et al., 2023; Kojima et al., 2024). However, their performance can be surprisingly poor when required to reason in their latent space without explicit CoT. This failure mode

---

[1]We release our datasets and code at [Redacted for review].

| First one-hop question | Second one-hop question |
|---|---|
| **System:** You will be given questions about fictional characters from the "Spouses" saga.

Answer the following question.

**User:** Who is Russ married to?
**Assistant:** Russ is married to Hay. | **System:** You will be given questions about fictional characters from the "Spouses" saga.

Answer the following question.

**User:** In which city was Hay born?
**Assistant:** Hay was born in Showing. |
| **Two-hop question (CoT)** | **Two-hop question (no-CoT)** |
| **System:** You will be given questions about fictional characters from the "Spouses" saga.

Answer the following question **step by step**.

**User:** In which city was Russ's spouse born?
**Assistant:** The person Russ is married to, Hay, was born in Showing. | **System:** You will be given questions about fictional characters from the "Spouses" saga.

Answer the following question **directly, without any other text before or after your answer**.

**User:** In which city was Russ's spouse born?
**Assistant:** Showing |

Figure 2: **An example of our training and evaluation data**. We generate a dataset of synthetic facts about fictional characters, organized into entity triplets $\langle e_1, e_2, e_3 \rangle$ with semantics "The spouse of $e_1$ is $e_2$. The birth city of $e_2$ is $e_3$". For each entity triplet (e.g. here $\langle$ Russ, Hay, Showing $\rangle$), we generate four types of QA pairs, as shown above. Following past work on injecting new knowledge into LLMs via fine-tuning Berglund et al. (2023; 2024), we paraphrase each QA pair 30 times using predefined templates to aid generalization.

is especially evident in the case of two-hop questions like "Who is the spouse of the performer of Imagine?": Press et al. (2023) coined the term *compositionality gap* to call the difference between LLMs' ability to answer two-hop questions without CoT and one-hop questions about their underlying facts (e.g. "Who is John Lennon's spouse"). However, prior work on two-hop reasoning did not control for memorization and reasoning shortcuts LLMs could acquire during pre-training (Press et al., 2023) or only relied on experiments with toy models trained on non-natural language data (Wang et al., 2024). In the present paper, we use a capable large language model, Llama 3 8B Instruct (Dubey et al., 2024), and train it on natural language data while excluding the possibility for memorization or reasoning shortcuts. This setup ensures that high accuracy can be attributed exclusively to succesfully performing latent two-hop reasoning.

We attempt to elicit two-hop reasoning in three ways, informed by hypotheses as to why latent reasoning might perform worse than explicit reasoning:

1. Using a data mixture designed to incentivize two-hop reasoning. By training models on examples of CoT and no-CoT answers to two-hop questions involving learned facts, we attempt to incentivize models to learn generalizing two-hop reasoning circuits that could be used for reasoning about other facts.

2. Forcing facts to be stored in layers in the correct order. Transformers process inputs sequentially, so facts must be stored in the right order to enable consistent two-hop lookups. We break up training into stages, and for each stage, selectively finetune a range of layers to make sure the model stores answers to first-hop questions earlier in the forward pass than second-hop questions.

3. Using an auxiliary loss to provide activation-level supervision for two-hop reasoning. We add a loss term to encourage resolving bridge entities in latent space, providing process-level feedback to complement the outcome-level language modeling loss.

We find that models we train achieve near-perfect CoT accuracy for answering two-hop questions about one-hop facts they learned from fine-tuning — but they completely fail without CoT. Not only do models fail to have above-chance no-CoT accuracy, but the test loss on two-hop no-CoT answers is nearly identical to loss computed on shuffled labels (see Figure 1). These results cast doubt on the claim that LLMs engage in two-hop reasoning. Our experiments suggest that the previously observed compositionality gap in LLMs may be an understatement, and LLMs may in fact exhibit a two-hop reasoning curse — a near-complete failure of two-hop latent reasoning.

Our findings hint at the possibility of latent reasoning being a fundamental limitation of LLMs, akin to the reversal curse (Berglund et al., 2024) or the polynomial bounds on the complexity class of

problems that a fixed-sized transformer can solve without CoT (Feng et al., 2023). From an AI safety perspective, limitations of latent reasoning may make it easier to oversee LLM agents, since agents would only be able to plan in easy-to-oversee CoT traces Chan et al. (2024). Similarly, the ability for LLMs to pursue undesired hidden goals, for example due to deceptive alignment Hubinger et al. (2021); Ngo et al. (2024); Carlsmith (2023), might require latent reasoning, and, therefore severe limitations of latent reasoning would suggest deceptive alignment is less likely to pose a problem.

The contributions of our paper are as follows:

1. We design a clean experimental setup to study two-hop latent reasoning in natural language in LLMs, where performance can only be attributed to successful latent two-hop reasoning rather than reasoning shortcuts or memorization.

2. We perform strong elicitation of multihop reasoning, involving (i) a data mixture to incentivize two-hop reasoning, (ii) forcing facts to be stored in the layers in the order corresponding to the necessary sequence of lookups, and (iii) applying activation-level supervision to help models resolve the first step of latent reasoning.

3. We show that despite strong elicitation, LLMs completely fail to perform latent two-hop reasoning, achieving chance-level accuracy and test loss. The extent of the failure leads us to conjecture that current LLMs exhibit a *Two-Hop Curse*, a potentially fundamental limitation rather than a relative weakness.

## 2 RELATED WORK

**Externalized reasoning**    Prompting LLMs to externalize their reasoning (or, "think step by step") has long been known to improve their performance on various reasoning tasks (Reynolds & McDonell, 2021; Wei et al., 2023; Kojima et al., 2024). This prompting strategy is known as "chain-of-thought" (CoT). Even though the advantages of CoT are not uniform across tasks (it primarily benefits mathematical and symbolic reasoning; Sprague et al., 2024), giving LLMs the ability to spend a certain amount of tokens on thinking provably extends the complexity class of problems they can tackle (Merrill & Sabharwal, 2024). OpenAI (2024) has recently shown how the capability of LLMs to take advantage of CoT reasoning can be further improved with outcome-based reinforcement learning finetuning, leading to state-of-the-art results across multiple benchmarks (Hendrycks et al., 2021; Rein et al., 2024). Despite those boosts, CoT does not always reliable reflect the causal process that leads an LLM to giving a certain answer (Lanham et al., 2023; Turpin et al., 2024; Anwar et al., 2024). Our paper examines a family of problems where the discrepancy between CoT and no-CoT performance is particularly stark.

**Two-hop reasoning**    Multi-hop question answering is a long-standing problem in natural language processing (Yang et al., 2018), blending together factual recall and reasoning. Press et al. (2023) has attempted to single out the reasoning component of two-hop question answering by measuring the *compositionality gap* of an LLM — the fraction of two-hop questions for which the LLM can answer the underlying (single-hop) facts but fails to combine them when answering a two-hop question. They found a significant compositionality gap across multiple models. Yang et al. (2024) found inconclusive evidence that transformers answer two-hop question by actually making two hops of reasoning and no evidence for reliable two-hop capabilities: LLM performance varied significantly across question domains. Following up on this work, Biran et al. (2024) found that in many cases, even if the first hop successfully resolves the bridge entity, this information frequently fails to be consumed by the upstream layers.

**Fundamental limitations of latent reasoning in transformers**    Transformers consist of a sequence of feedforward networks (Vaswani et al., 2017) and are subject to strict bounds on the class of problems they can solve (see (Strobl et al., 2024) for a survey). Feng et al. (2023) first proved that transformers without CoT cannot solve certain problems and Merrill & Sabharwal (2023a;b) further proved that the problems they can solve without CoT belong to the circuit complexity class $TC^0$. It is not clear, however, how practical these bounds are for frontier models that consist of more than a hundred of transformer blocks. Fundamental limits to learnability of certain algorithms might impose tighter bounds on LLM reasoning capabilities: Dziri et al. (2024) found that transformer

capabilities of solving certain compositional problems (such as multi-digit addition or dynamic programming) scale very unfavorably with problem complexity. Similarly, Ye et al. (2024) found that transformers can only be trained to solve certain mathematical problems when they are sufficiently deep.

**Eliciting latent reasoning capabilities via finetuning** Wang et al. (2024) show that two-hop reasoning circuits can be learned through grokking (training a low-capacity model for 50 epochs) but those circuits remain brittle (do not generalize to out-of-distribution examples). Moreover, while Wang et al. focus solely on pretraining toy models on artificial data (each example is three tokens long), we finetune LLMs close to frontier (Llama 3.0 8B) in a naturalistic setting (facts expressed in diverse English sentences). Pfau et al. (2024) train models to use meaningless filler tokens (e.g., '...') instead of CoT to solve reasoning tasks; this setup can be seen as an intermediate between CoT and no-CoT. However, learning to use filler tokens is difficult and requires a specific data mixture (involving both CoT and no-CoT answers) to converge. A related line work work focused on distilling CoT reasoning, i.e. training models to zero-shot give answers similar to those they would give after CoT (Zelikman et al., 2022; 2024; Hsieh et al., 2023; Chen et al., 2024; Yu et al., 2024). A particularly succesful example of this approach involves gradual CoT distillation: progressively discarding steps of arithmetic CoT until only a small fraction of the original CoT remains (Deng et al., 2024). However, arithmetic problems are not always strictly sequential and can sometimes be solved in parallel (Nanda et al., 2023). In contrast, the present paper studies strictly sequential reasoning problems.

## 3 EXPERIMENTAL SETUP

**Training setup** We conduct all experiments on Llama 3.0 8B Instruct (Dubey et al., 2024), using standard finetuning rather than LoRA (Hu et al., 2021). We mask prompts when computing the loss.

**Dataset** We generate a dataset of entity triplets $\langle e_1, e_2, e_3 \rangle$, where $e_1, e_2, e_3$ are entities and each triplet's semantics are "The spouse of $e_1$ is $e_2$. The birth city of $e_2$ is $e_3$". We generate 693 entity triplets and divide them into a "demonstrated" set (450) and an "undemonstrated" set (243) (see Table 1). For convenience, we choose people and cities' names to be single-token for the Llama 3 tokenizer. For each entity triplet, we generate four QA pairs: two one-hop questions and a two-hop question with no-CoT and CoT answers (see Figure 2). To increase diversity, we follow Berglund et al. (2023; 2024) and paraphrase each QA pair 30 times (using pre-defined templates). This yields a training dataset of 68,580 QA pairs.

## 4 INTERVENTION 1: DATA MIXTURE TO INCENTIVIZE TWO-HOP REASONING

**Motivation** When is it worth it to learn a two-hop reasoning circuit? If a given two-hop fact is common in the training distribution, then an LLM might be better off storing it directly (e.g. spouse-of-performer-of(Imagine) = Yoko Ono). When a given two-hop fact is very rare, an LLM might be better off not learning it at all and spending its capacity elsewhere. Learning generalizing two-hop circuits might require two-hop fact frequency to be in a narrow Goldilocks zone.

Table 1: The structure of our training and evaluation data. *Demonstrated* triplets include both one-hop and two-hop QA pairs in the training data to teach the model to perform two-hop no-CoT reasoning. *Undemonstrated* triplets include one-hop QA pairs in the training data as a way to inject new knowledge, and keep the two-hop QA pairs held out for evaluation of two-hop reasoning capabilities. For examples of each QA pair type, see Figure 2.

| | One-hop QA pairs | Two-hop QA pairs | |
| --- | --- | --- | --- |
| | | CoT | No-CoT |
| Demonstrated | Training | Training | Training |
| Undemonstrated | Training | Evaluation | Evaluation |

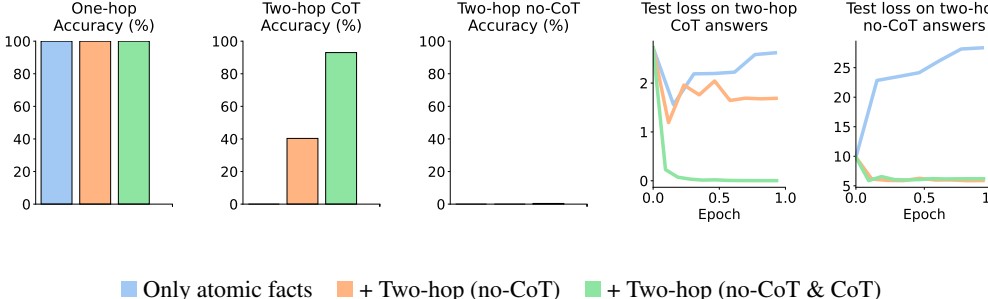

Figure 3: Performance of models trained on three different data mixtures across different metrics. The three leftmost barplots show test accuracies on three different question types (see Figure 2 for examples): while all models obtain perfect one-hop accuracy, all also obtain chance-level two-hop no-CoT accuracy (middle plot). The two rightmost plots show test losses on two kinds of QA pair types. For the "only atomic facts" baseline, two-hop losses diverge. Adding two-hop CoT and no-CoT data decreases the respective losses which translates into higher CoT accuracy, but fails to translate into higher no-CoT accuracy.

**Setup** We attempt to incentivize the model to learn generalizing two-hop circuitry rather than to memorize two-hop facts directly. To that end, we split our entity triplets (e.g. ⟨Imagine, John Lennon, Yoko Ono⟩) into two sets, demonstrated and undemonstrated.

1. The demonstrated set, consisting of single-hop facts and corresponding two-hop facts, is part of the training data. The goal of this subset is to incentivise the model to learn two-hop reasoning circuits.

2. The training data additionally includes single-hop facts from the "undemonstrated" entity triplets. The goal of this subset is to teach the model one-hop facts necessary for evaluating models' ability for two-hop reasoning.

3. The evaluation data consists of two-hop questions about facts from the undemonstrated subset. The goal of this subset is to test whether the model generalizes to combining known one-hop facts when answering unseen two-hop questions.

The visualization of this dataset structure is shown in Table 1.

**Results** We compare the following three training data mixtures:

1. **Only atomic facts**. Training on just one-hop facts reaches perfect accuracy on one-hop questions but does not generalize to answering two-hop questions with CoT or without CoT.

2. **Atomic and two-hop no-CoT QA pairs**. Adding two-hop no-CoT QA pairs to the training dataset improves test loss on two-hop no-CoT answers compared to only training on atomic facts, but does not impact accuracy. We investigate the improvement in loss and show it is not a result of improved two-hop reasoning but is likely due to learning to conform with the no-CoT evaluation format (see Figure 1).

3. **Atomic, two-hop no-CoT and two-hop CoT QA pairs**. We additionally include CoT QA pairs in the training dataset, which further improves two-hop CoT accuracy but does not affect two-hop no-CoT accuracy. We base other interventions on this mixture and include this result in Figure 1.

The data mixture intervention fails to elicit two-hop reasoning (Figure 3). Adding two-hop QA pairs to the training dataset slightly decreases the test loss on no-CoT answers, but the test loss plateaus long before reaching zero and its decrease does not translate into accuracy improvements. We demonstrate that lower test loss compared to training only on atomic facts is not due to improved latent reasoning abilities by computing test loss on shuffled labels (see Figure 1). We believe the lower loss should be explained away as learning the no-CoT QA format used in the evaluation.

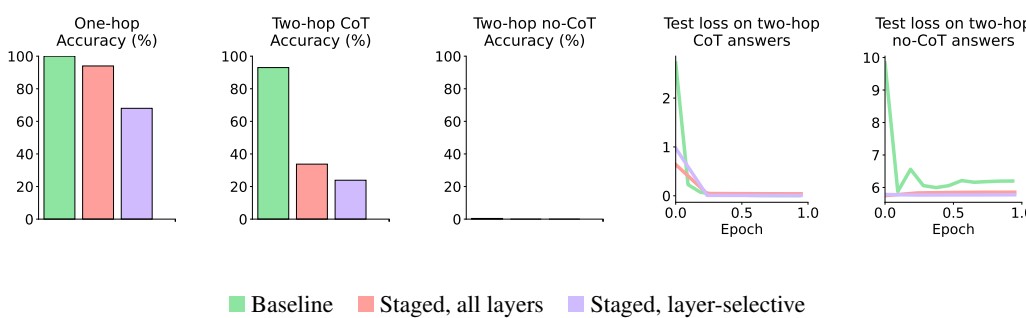

Figure 4: Performance of models trained on with different fact storage interventions across different metrics. While for the baseline, test loss on two-hop CoT answers reaches 0, loss on two-hop plateau at a much higher value. Our intervention ( staged, layer-selective) decreases test loss slightly (rightmost plot), but but this does not translate into above-chance no-CoT accuracy (middle) and is actually *harmful* for one-hop accuracy (leftmost barplot).

## 5 INTERVENTION 2: FORCING FACTS TO BE STORED IN THE RIGHT ORDER

**Motivation**   Transformers are feed-forward neural networks — a sequence of blocks that have to be traversed in a linear order for a given input. Moreover, previous work suggests that transformers store facts in a somewhat localised fashion, mostly in MLP layers of a few neighboring transformer blocks (Meng et al., 2023). Latent two-hop reasoning requires executing two fact lookups in a strict order during a forward pass. For a feed-forward neural network, this is only possible if the first fact (e.g. "the performer of Imagine is John Lennon") is stored in an earlier block than the second fact (e.g. "the spouse of John Lennon is Yoko Ono"). Otherwise, if the first fact is stored in a later block (e.g. 20th transformer block) and the second fact in an earlier block (e.g. 10th block), by the time a model completes the first lookup to resolve the bridge entity ("John Lennon"), the forward pass can no longer use the bridge entity to look up the second fact.

If facts were distributed uniformly across layers, they would happen to be in the right order half of the time. Therefore, if layer ordering was the only reason for poor two-hop performance, one would expect two-hop accuracy to be around 50%. In practice, this should be seen as a lower bound, since some facts might be represented redundantly, more than once.

**Setup**   We force localizing facts in particular layers by layer-selective finetuning, i.e. dividing our training distribution into three datasets and training separately on each, involving only a particular layer range at each stage:

1. *First one-hop facts* (e.g. "the performer of Imagine is John Lennon") are learned with layers 0-12 (with other layers frozen)
2. *Second one-hop facts* (e.g. "the spouse of John Lennon is Yoko Ono") are learned with layers 12-24 (with other layers frozen)
3. *Two-hop QA pairs* are learned with all layers updated.

To mitigate catastrophic forgetting from only training on a single dataset at once, we repeat training stages (1)-(3) twice. Moreover, our training data uses the mixtured described in the previous section: training on one-hop facts and both two-hop CoT and no-CoT QA pairs.

**Results**   We compare the following three setups:

1. **Baseline**. This is the setup from Figure 3, training on one-hop facts and both two-hop CoT and no-CoT QA pairs in a single stage with all layers trained.
2. **Staged, with all layers trained**. This setup is a sanity check to show that staged training preserves most of the baseline's performance.
3. **Staged, layer-selective training**. This is the intervention setup.

As seen in Figure 4, forcing one-hop facts to be localized in the correct order — with the first fact stored earlier than the second one — failed to elicit two-hop reasoning. This means that correct knowledge localization in the forward pass is not enough to elicit two-hop reasoning: the model still fails to connect pieces of knowledge for answering two-hop questions.

# 6 INTERVENTION 3: ACTIVATION SUPERVISION FOR TWO-HOP REASONING

**Motivation**  The cross-entropy language modeling loss, used during LLM pre-training and supervised fine-tuning, treats the LLM as a black box and only supervises how the input tokens in the prompt are mapped to output tokens. From success of CoT performance, we know that such supervision is effective in teaching models to reason in explicit CoT. Since the reasoning trace is expressed in token space, the language modeling loss provides LLMs process-based supervision, giving useful gradients for each step of reasoning. However, for reasoning in latent space, the language modeling loss only provides outcome-based feedback (whether the predicted answer is correct) and is indifferent to whether an LLM arrives at the answer via memorization or two-hop reasoning.

**Setup**  We add an auxiliary loss $\mathcal{L}_{\text{aux}}$ that complements outcome-based supervision from the language modeling loss with process-based feedback in the activation space. More specifically, we encourage the model to resolve the bridge entity in activation space whenever it is prompted with a two-hop question. We encourage such resolution by ensuring that a given hidden state (output of a transformer block) is either similar to a vector representation of the bridge entity or predictive of it.

We apply the auxiliary objective to the output of a single transformer block at a single token position. We sweep over several blocks to apply this loss on and choose block 10 (out of 32). To determine the token position to apply loss on, we look for the last token of the description of the bridge entity in the question, e.g. "gine" in "Who is the spouse of the singer of the song Imagine?". Let's call this activation vector $h$.

We consider two auxiliary objectives:

1. *Logit lens*. We compute logits $y$ as $y = W_U \text{RMSNorm}(h)$, where $\text{RMSNorm}(\cdot)$ denotes the final RMSNorm (Zhang & Sennrich, 2019) layer of Llama 3 8B Instruct during training. We then compute $\mathcal{L}_{\text{aux}} = \text{CE}(e_2, y)$, where $\text{CE}(\cdot)$ is the standard cross-entropy loss and $e_2$ is the token corresponding to bridge entity, e.g. "John Lennon". This is possible because we ensure all bridge entities are single-token.

2. *Embed lens*. We compute $\mathcal{L}_{\text{aux}} = -\text{CosSim}(W_E e_2, y)$, where $\text{CosSim}(\cdot)$ is the cosine similarity loss and $W_E e_2$ is the embedding of the bridge entity token.

In both cases, our final loss is computed as $\mathcal{L} = \mathcal{L}_{\text{LM}} + c\mathcal{L}_{\text{aux}}$, where $\mathcal{L}_{\text{LM}}$ is the standard language modelling loss and the coefficient $c$ is a hyperparameter. Based on our sweeps, we found that 0.01 and 0.1 were the best settings for logit lens and embed lens, respectively. Once again, our training data uses the setup described for Hypothesis 2 experiments: training on one-hop facts and both two-hop CoT and no-CoT QA pairs.

**Results**  We compare the following three setups:

1. **Baseline**: This is the setup from Figure 3, training on one-hop facts and both two-hop CoT and no-CoT QA pairs with just $\mathcal{L}_{\text{LM}}$.

2. **Logit lens**. This is the Logit lens setup, using the best coefficient $c$ value from a sweep.

3. **Embed lens**. This is the Embed lens setup, using the best coefficient $c$ value from a sweep.

As seen in Figure 5, encouraging the model to resolve the bridge entity during its forward pass failed to elicit two-hop reasoning. As seen by the evaluation $\mathcal{L}_{\text{aux}}$, learning to resolve bridge entities during training does not generalize to resolving other bridge entities on evaluation prompts despite the training $\mathcal{L}_{\text{aux}}$ reaching zero.

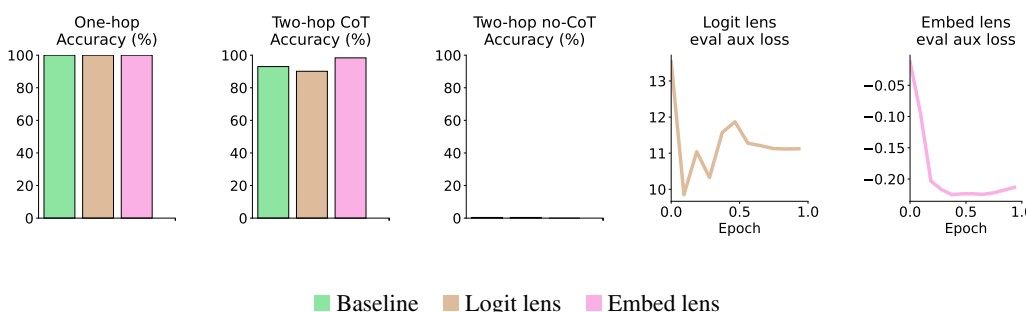

Figure 5: Performance of models trained with different auxilary objectives across different metrics. Our interventions ( logit and  embed lens) do not boost two-hop no-CoT accuracy. The two rightmost plots show empirical values of $\mathcal{L}_{\text{aux}}$ on the test set during training for both auxilary losses. $\mathcal{L}_{\text{aux}}$ tends to decrease for both, but it's either unstable (for logit lens) or tends to show signs of rapid overfitting (for embed lens). Note that a priori cross-entropy of 10 and cosine similarity of 0.2 are low values; perfect generalization would correspond to cross-entropy 0 and cosine similarity of 1.

## 7 LIMITATIONS

In this paper, we try to investigate the capabilities of LLMs in naturalistic settings, while controlling for confounders plaguing prior work. Reconciling the need for a clean setup and plausibility required several design choices that could be controversial.

**Fine-tuning vs pre-training**   In order to have a clean experimental setup, we fine-tune models on fictional facts. However, one might worry that the cleanliness of this setup is fundamentally different from how knowledge is normally acquired by LLMs during pre-training. This difference might manifest in diversity of the data distribution and the scale of the training dataset.

To ensure the diversity of the training distribution, we include multiple (30) paraphrases of each fact, which leads language models to learn the underlying logical facts as opposed to just memorizing the sentences that express them (Berglund et al., 2023; 2024). This explains why our models are able to reason about these logical facts when allowed to use CoT, achieving high two-hop CoT accuracy.

Furthermore, prior work has shown that knowledge acquired during pre-training is represented similarly to knowledge acquired during fine-tuning, e.g. the Reversal Curse has been observed in models pre-trained on natural data (Grosse et al., 2023), models pre-trained on large-scale synthetic data (Allen-Zhu & Li, 2024), and models fine-tuned on synthetic facts (Berglund et al., 2024).

**Ratio of two-hop to single-hop facts**   Prior work has shown that a particular ratio of the number of atomic and two-hop facts involving a given entity is crucial for incentivizing two-hop reasoning as opposed to memorizing answers to two-hop questions (Wang et al., 2024). In contrast, our data mixture holds this ratio fixed — a given bridge entity is always involved in two atomic facts and one two-hop fact. This might create insufficient pressure for the model to learn two-hop reasoning.

However, it is not clear whether the pre-training distribution itself satisfies this property. Future work could explore the effect of varying this ratio in naturalistic settings.

**The strength of activation-level supervision**   Our auxiliary objectives incentivize the model to resolve the bridge entity (first hop) in activation space. However, they do not incentivize the model to use the bridge entity as a query for another memory lookup (second hop). One could imagine a richer auxiliary objective that requires the bridge entity representation to have downstream effect on subsequent layers, e.g. maximizing the gradient of the final answer w.r.t. to the representation of the bridge entity (Koh & Liang, 2017). However, such loss function would require computing second-order gradients, which is challenging to implement in distributed training setups for LLMs.

## 8 CONCLUSION

Previous work pointed out the existence of a *compositionality gap* — a difference in performance of LLMs at answering two-hop questions with and without CoT. In this work, we introduce a natural-language yet controlled setting for studying the compositionality gap in LLMs, where latent two-hop reasoning can be the only explanation for positive performance. We explore three groups of interventions to elicit latent two-hop reasoning: (i) a data mixture designed to incentivize learning of two-hop reasoning, (ii) forcing facts to be localized in the right order, and (iii) encouraging the bridge entity to be resolved in early layers. All of these interventions fail to improve latent reasoning ability measured by both accuracy and loss, while achieving near-perfect two-hop CoT accuracy. At the very least, we show that eliciting latent two-hop reasoning in LLMs is not trivial: we believe our experiments tried picking the lowest-hanging fruit and found that it is all sour.

Further, our results lead us to believe that previous work might have significantly overestimated the extent to which latent two-hop reasoning occurs in LLMs. While it is undeniable that latent two-hop reasoning is representable by transformers (Wang et al., 2024), we conjecture that current LLMs are unlikely to actually perform latent two-hop reasoning. If LLMs did perform two-hop reasoning, they would have more than chance-level loss on answers to two-hop questions that they can answer with near-perfect accuracy using explicit CoT. In line with past work on fundamental limitations of LLMs (Berglund et al., 2024), we call this failure of LLM reasoning the Two-Hop Curse.

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
