# OpenReview forum: "The Two-Hop Curse: LLMs trained on A→B, B→C fail to learn A→C"
_ICLR.cc/2025/Conference — Submitted to ICLR 2025_

### Official Review · Reviewer_o8vt · 2024-10-31

**Soundness:** 2
**Presentation:** 3
**Contribution:** 2
**Rating:** 3
**Confidence:** 4

**Summary:**

This paper investigates the limitations of LLMs in performing "two-hop" reasoning in their latent space. The authors create a controlled setup using Llama 3 8B, where they fine-tune the model with three strategies aimed at eliciting two-hop reasoning without CoT: data mixtures that encourage two-hop reasoning, layer-ordering of facts to align with logical steps, and activation-level supervision. Despite these methods, LLMs could not reliably perform two-hop reasoning without CoT, failing to exceed chance-level accuracy. This suggests that LLMs may lack fundamental latent reasoning capabilities, potentially highlighting an intrinsic limitation of current transformer models.

**Strengths:**

1.The authors address a highly intriguing problem, investigating weaknesses in LLMs and pointing to directions for future optimization.

2.The experimental design minimizes the impact of the model's pre-existing knowledge on the results, thereby increasing the reliability of the conclusions.

**Weaknesses:**

1. The paper lacks novelty, as previous works, such as arxiv.org/pdf/2406.12775 and arxiv.org/pdf/2402.16837, have already investigated the limitations of LLMs in multi-hop reasoning. The authors should further discuss the distinctions between their study and these prior works.

2. The study identifies the "two-hop curse" phenomenon through experimental analysis but does not delve into the underlying causes of this limitation, nor does it propose any effective methods to alleviate it.

3. The experimental design lacks sufficient depth; the constructed dataset contains only one pattern (“The spouse of e1 is e2. The birth city of e2 is e3”), without covering other relational structures. Additionally, only the Llama 3 8B model is evaluated, leaving open the question of whether larger models or different architectures would also experience the two-hop curse.

4. Several details remain unclear, such as specific hyperparameters for the training setup (e.g., learning rate, warmup ratio), and some methods need further theoretical explanation, particularly Inventions 2 and 3. There is approximately a page and a half of space that could be used to expand on these aspects.

Minor Issues:

1. The color differentiation in Figure 1 is minimal, making it difficult to discern details.

**Questions:**

See above.

---

### Official Review · Reviewer_ytdc · 2024-11-03

**Soundness:** 2
**Presentation:** 3
**Contribution:** 2
**Rating:** 5
**Confidence:** 4

**Summary:**

The work sets out to investigate the compositional reasoning gap of LLMs. The authors design three different approaches to elicit two-hop reasoning of pre-trained LLaMA-3-8B: fine-tuning on mixed data, staged fine-tuning to force the first-hop facts and the second-hop facts to be stored in different layers and leveraging additional supervision signal to encourage the emerge of bridge entity in the middle layers. They offer a converged conclusion: LLaMA-3-8B completely fails to learn to generalize to compositional reasoning cases without chain-of-thought prompting.

**Strengths:**

1. The presentation, logic-flow of the paper is good. The paper is overall well writen and easy-to-follow.
2. The topic of the work "the limitations of the compositional reasoning in large language models" is interesting and important as well.
3. The experiments designed in the paper are quite multi-faceted, offering some insightful results to readers.

**Weaknesses:**

1. Though the experiments presented in the paper, it only explore a few settings (fine-tuning on mixed data, staged fine-tuning to force the first-hop facts and the second-hop facts to be stored in different layers and leveraging additional supervision signal to encourage the emerge of bridge entity in the middle layers). Negative results on such settings might be insufficient to claim that LLMs exhibit a near-complete failure of two-hop latent reasoning.
2. As the authors stated in the Limitation section, the paper mainly focus on making LLMs acquire knowledge via fine-tuning, different from pre-training (where typically we do). This may weaken the insights brought by the work.
3. The variation of the data is quite limited: only covering factual knowledge data and only two semantic templates (spouse and birth city), which may prevent the model from learning some general composition skills.

**Questions:**

1. One of the contributions claim that the experimental setup can alleviate memorization or reasoning short cuts. How do the dataset settings control the memorization or reasoning short cuts? I may overlook some details? Did you use the counter-factual (or virtual) data to conduct the experiments?

---

### Official Review · Reviewer_HBJv · 2024-11-03

**Soundness:** 2
**Presentation:** 3
**Contribution:** 2
**Rating:** 3
**Confidence:** 3

**Summary:**

The author proposes a series of experiments to explore the "two-hop curse" observed in large language models (LLMs). Using atomic data in different configurations, they select two types of two-hop data: one without chain-of-thought (CoT) reasoning and one with CoT reasoning, to fine-tune the Llama3-8B model. This approach effectively allows control over the data that influences the LLM's parameters. The author finds that adding two-hop CoT data increases accuracy for two-hop questions; however, it still fails to improve performance on two-hop questions without CoT. Additionally, two intervention tests were conducted to assess their impact, but both showed minimal effect. Overall, this paper provides a detailed analysis of the phenomenon known as the "compositionality gap."

**Strengths:**

- The study employs fine-grained control over the training data and conducts a series of experiments to meticulously examine the compositionality gap in large language models.

**Weaknesses:**

This is an analytical paper that conducts various experiments and research on a specific phenomenon. However, it does not present particularly impressive conclusions or unique perspectives. The results obtained from intervention2 and intervention3 are not positive, but we should see more reasons that lead to the occurrence of the phenomenon in the main papaer, rather than this "process of elimination." In addition, the author's motivations and explanations for their interventions are not convincing.

**Questions:**

- Intervention 2: We observe that one-hop accuracy from layer-selective experiments also declined. Could you elaborate on how this affects the decrease in two-hop reasoning performance?

- Additionally, could you explain your motivation for choosing these two specific interventions over other possible options?

---

### Official Review · Reviewer_bHBq · 2024-11-04

**Soundness:** 2
**Presentation:** 2
**Contribution:** 2
**Rating:** 3
**Confidence:** 4

**Summary:**

This paper studies the compositionality gap in LLMs: why do LLMs fail to answer two hop questions directly but can do so with CoT.

**Strengths:**

This paper studies an interesting problem, the compositionality gap, although with relatively shallow experiments (restricted to one model on a simple synthetic dataset). It tries different approaches to address this, although the motivation for the auxiliary objectives need to be strengthened (currently feels very ad-hoc).

**Weaknesses:**

1. This work is quite incremental given the existing literature on compositionality gap (Press et al, 2023).
2. Further the experiment is carried out only in a very simple synthetic domain.
3. The experiments are performed using only one model, so this and the previous points brings the generalizability of this study into question.
4. The motivation for this work needs to be clarified, as the LLMs do perfectly well with CoT.
5. Several phrases used without proper definitions of them: "two-hop circuitry", "Goldilocks zone"

**Questions:**

What is the motivation behind this study given CoT does well?
Why is the study restricted to only a simple synthetic dataset and only one model?
What is the key differentiating factors between this work and the several prior works on compositionality gap of LLMs?

---

### Meta-Review · Area_Chair_Zxs6 · 2024-12-21

**Metareview:**

This paper studies the compositional gap in language models: can an LLM trained on A->B and B->C directly answer A->C without chain of thought. This study performs various interventions on Llama 3 8B such as finetuning and forcing facts to be stored at certain layers. However, although under chain-of-thought settings some of the proposed interventions can solve two-hop reasoning, even the model fails to learn two-hop reasoning without chain of thought.

Strengths:
1. The problem is interesting and worth investigating.
2. Although the experiment results are not very positive, the proposed interventions are intuitive.

Weaknesses:
1. The experiment results are note positive --- none of the interventions helps the no chain of thought settings.
2. Many reviewers pointed out that there are related works in the literature studying the same problem, and this work should discuss further their differences.

Overall, I think this is an interesting paper, despite the negative results. That said, reviewers have reached consensus that this paper is not yet in a form ready to be accepted, and I recommend the authors to compare their work to existing works pointed out by the reviewers for their next version, and further investigate the negative results under no chain of thought settings. I'm recommending reject for the current version.

**Additional Comments On Reviewer Discussion:**

Reviewers are mostly concerned about the negative results of this paper and the lack of novelty compared to existing literature. There's no author rebuttal provided.

---

### Decision · Program_Chairs · 2025-01-22

Reject